# A Microarray, Validation, and Gene-Enrichment Approach for Assessing Differentially Expressed Circulating miRNAs in Obese and Lean Heart Failure Patients: A Case–Control Study

**DOI:** 10.3390/ijms26199475

**Published:** 2025-09-27

**Authors:** Douglas dos Santos Soares, Amanda Lopes, Mariana Recamonde-Mendoza, Rodrigo Haas Bueno, Raquel Calloni, Nadine Clausell, Santiago Alonso Tobar Leitão, Andreia Biolo

**Affiliations:** 1Experimental and Molecular Cardiovascular Laboratory, Heart Failure and Cardiac Transplant Unit, Cardiology Division, Hospital de Clínicas de Porto Alegre, Porto Alegre 90035-903, Brazil; soaresdouglas.ef@gmail.com (D.d.S.S.); alopestk@gmail.com (A.L.); nclausell@hcpa.edu.br (N.C.);; 2Postgraduate Program in Cardiology and Cardiovascular Science, Federal University of Rio Grande do Sul, Porto Alegre 90040-060, Brazil; 3Institute of Informatics, Universidade Federal do Rio Grande do Sul (UFRGS), Porto Alegre 90035-007, Brazil; mari.mendoza@gmail.com; 4Bioinformatics Core, Hospital de Clínicas de Porto Alegre (HCPA), Porto Alegre 90035-003, Brazil; 5Federal Institute of Education, Science and Technology of Rio Grande do Sul—Campus Gravataí, Gravataí 92412-240, Brazil

**Keywords:** heart failure, obesity, microRNA, circulating microRNA, microarray, bioinformatic

## Abstract

Obesity is a risk factor associated with cardiovascular diseases that may lead to heart failure (HF). However, in HF, overweight and obese patients have longer survival than underweight patients, a phenomenon known as the obesity paradox. MiRNAs play a fundamental role in gene regulation involved in obesity and HF. The main objective of this study was to identify and validate differentially expressed circulating miRNAs in HF–obese and HF–lean patients. This case–control study was carried out in two phases: discovery and validation. In the discovery phase, plasma samples from 20 HF patients and from 10 healthy controls were analyzed using the miRNA 4.0 Affymetrix GeneChip array. Differentially expressed miRNAs were ranked and selected for validation. In this phase, plasma miRNAs -451a, -22-3p, and -548ac from 80 patients and controls were analyzed by qPCR. Target analysis and functional enrichment analysis were performed. When comparing HF–lean and HF–obese groups compared to controls, miRNAs -451a and -22-3p were up-regulated in both discovery and validation phases, while -548ac was down-regulated in the discovery phase and up-regulated in the validation phase, indicating that miRNA changes are independent of obesity. These miRNAs regulate genes and different biological processes associated with metabolic, morphological, and functional outcomes.

## 1. Introduction

Obesity is one of the main risk factors associated with cardiovascular diseases that culminate in heart failure (HF) [1,2,3,4]. The higher the body mass index, the greater the risk of developing HF [3,4]. However, once HF is established, overweight and obese patients have more prolonged survival than patients with low weight and severe obesity [5,6,7,8,9]. This paradox has drawn the scientific community’s attention to better understanding the possible mechanisms involved in this scenario. MiRNAs play a key role in gene regulation, and there are transcription factors involved in obesity [10,11] and HF [11,12]. In a previous study from our group, the miR-221/-130b ratio was significantly associated with adiposity in patients with heart failure in comparison to healthy control participants [13]. Additionally, computational analysis showed that miR-130b and miR-221 regulate multiple interconnected pathways that are associated with endocrine and cardiovascular diseases [13]. Studying the mechanisms involved in this scenario is of great importance to better understand the role of obesity in HF.

Therefore, the main objective of this study was to identify and validate the differentially expressed miRNAs involved in the obesity paradox and to evaluate the mechanisms potentially modulated by their action.

## 2. Results

### 2.1. Discovery Cohort

#### 2.1.1. Baseline Characteristics

A total of 20 HF patients (10 obese and 10 lean) and 10 healthy controls were enrolled for the discovery phase. Clinical and laboratory characteristics of patients and controls are shown in Table 1. The three groups were balanced for age, sex, and ethnicity. The obese HF group presented the adiposity parameters characteristic of the population with increased body weight, BMI, fat percentage, and abdominal circumference. Both HF groups had severe systolic dysfunction and mild functional limitation. Most patients were under a standard therapeutic regimen with beta-blockers and angiotensin-converting enzyme inhibitors.

#### 2.1.2. Microarray Results

In the microarray analysis, from the 3770 probesets present in the chip, a total of 2578 human miRNAs were detected; 17 miRNAs were found to be differentially expressed in the comparison between HF–lean and controls, 19 miRNAs in the comparison between HF–obese and healthy controls, and 2 miRNAs in the comparison between HF–obese and HF–lean. Afterward, we built a Venn diagram to evaluate the overlaps of each comparison. Eleven HF-related miRNAs and one obesity-related miRNA were observed. Among the HF-related miRNAs, we observed up-regulated miR-22-3p, miR-378h, miR-106b-5p, miR-140-3p, miR-6125, and miR-3196, and down-regulated miR-548ac, miR-3128, miR-3201, miR-574-5p, and miR-8084. Obesity-related miR-451a was up-regulated (Figure 1; Appendix A).

### 2.2. Validation Cohort

#### 2.2.1. Baseline Characteristics

A total of 80 subjects, 61 HF patients (26 obese and 35 lean) and 19 healthy controls, were enrolled for the validation phase of the study. Clinical and laboratory characteristics of patients and controls are shown in Table 2. The obese HF group was well characterized for obesity parameters. Both HF groups had severe systolic dysfunction and mild functional limitation. Most patients were under a standard therapeutic regimen with beta-blockers and angiotensin-converting enzyme inhibitors. There were no significant differences in clinical variables between both HF groups except for hypertension.

#### 2.2.2. Validation Results

Based on microarray results, we ranked miRNAs by effect size (−1; 1-fold change), lowest *p*-value (<0.01), and biological plausibility to prioritize candidates for validation. Subsequently, miRNAs -451a, -22-3p, and -548ac were selected for further analysis. Notably, both miR-451a and miR-22-3p were up-regulated in both HF groups compared to the control, validated in our sample. However, miR-548ac was down-regulated in the microarray and up-regulated in the validation sample, not validating the initial result (Figure 2, Appendix A).

#### 2.2.3. Construction of HF-miR–Gene Network

After identifying and ranking the differentially expressed miRNAs, we performed an interaction analysis among miRNAs–genes–pathways. The interaction of these miRNAs resulted in a network with 937 genes involved. We then filtered genes that interact with at least two different miRNAs (miR-451a; miR-22-3p; miR-548ac), which yielded a network with 36 genes (Figure 3). We selected the signaling pathways that interacted with these genes and filtered them by FDR < 0.05, totaling nine pathways (Appendix A). Finally, we assessed the most frequent genes across these pathways and ranked them. The most frequent genes were *AKT1, MAPK1, GRB2, IGF1R, PTEN, ESR1, HSPA1B, MAP3K1*, and *ZFHX3*. These genes are associated with different biological processes, such as apoptosis, carbohydrate metabolism, neurogenesis, sugar transport, translation regulation, transport, the cell cycle, actin cytoskeleton reorganization, cardiac neural crest cell development involved in heart development, and aging.

## 3. Discussion

In the discovery phase of the present study, of the 2578 human miRNAs, 38 were differentially expressed—11 related to HF and 1 to obesity. After evaluating the effect size, *p*-value, and biological plausibility of the differentially expressed miRNAs, we chose three of them for validation—miR-451a, miR-22-3p, and miR-548ac. In the validation phase, miRNAs -451a and -22-3p were up-regulated in the HF groups, regardless of obesity. miR-548ac was shown to be divergent between the discovery (down-regulated) and validation (up-regulated) phases in the HF groups compared to the control group (Figure 4). Through in silico analysis, we observed that this group of miRNAs is associated with some important biological processes that participate in metabolic, morphological, and functional activities in HF through genes such as *AKT1, MAPK1, GRB2*, and *IGF1R.*

Our results show that miR-451a is up-regulated in the HF groups regardless of obesity compared to the control group. Interaction analysis showed that miR-451a is associated with *AKT1*, *MAPK1*, and *MAP3K1* genes. These genes interact with biological processes, such as apoptosis, metabolism, proliferation, cell survival, growth, and angiogenesis, which are fundamental for cardiac remodeling and function. As in our population, patients with hypertrophic cardiomyopathy have increased levels of circulating miR-451a [14], showing an association with cardiac remodeling. Some preclinical studies have shown that the increased expression of miR-451a prevents activation of matrix metalloproteinases 2 and 9 in human cardiomyocytes during pathological stress stimulation [15] and attenuates cardiac fibrosis and angiotensin II-induced inflammation in mice [16]. In addition, down-regulation of miR-451a in the left atrium of patients with recurrent atrial fibrillation after ablation is associated with a higher percentage of atrial fibrosis and worse prognosis [17], reinforcing the cardioprotective role of this miRNA by modulating pathways related to cardiac remodeling and inflammation. Suppression of the miR-144/451 bicistronic gene locus plays an important role in the control of oxidative stress and erythrocyte production, which can have a crucial impact on the progression of heart damage to heart failure, and also in the worsening of heart failure symptoms [18].

Our analyses showed that miR-22-3p was up-regulated in both HF groups compared to the control group. Based on interaction analysis, the literature suggests that miR-22-3p is associated with *AKT1* and *MAPK1* genes, in conjunction with miR-451a, suggesting a synergy of these miRNAs in gene expression regulation and, consequently, modulation of the same biological processes. In the Bio-SHiFT study [19], it was observed that miR-22-3p is an independent marker, inversely associated with primary outcomes such as heart failure, hospitalization, cardiovascular mortality, cardiac transplantation, and LVAD implantation. A recent preclinical study [20] showed that up-regulation of miR-22-3p generated inhibitory effects on cell proliferation and collagen deposition in cardiac fibroblasts treated with Ang II. This evidence suggests that miR-22-3p is associated with cardioprotective factors that should be further explored.

The results of miR-548ac were divergent between the discovery and validation phases. While this miRNA was down-regulated in our microarray analysis, we observed its up-regulation in the RT-qPCR analysis, showing an increased expression in patients with HF compared to the control group. The miRNA target analysis highlighted genes *MAPK1*, *GRB2*, and *IGF1R* as potential targets for this miRNA. These genes participate in the activation of biological processes such as cell proliferation, cell surface growth factor receptors, the Ras signaling pathway, cell growth, and survival control. To date, our study appears to be the first to analyze miR-548ac in patients with HF. Previously, only studies evaluating its expression in the context of rheumatic diseases [21] and cancer [22,23,24] were reported. Our data suggest that exploring miR-548ac and the mechanisms involved in patients with HF may be relevant.

Although the rigorous methodological approach and balanced groups regarding key clinical variables provide valuable insights into the biological mechanisms underlying heart failure and obesity, particularly concerning circulating miRNAs, due to the exploratory nature of this study, there are several limitations to be acknowledged. First, no formal sample size calculation was performed, and the sample size was relatively small and based on convenience sampling, which may limit the statistical power to detect subtle differences and increase the risk of type II error. Therefore, our results do not allow any possible causal inference regarding miRNA regulation of the process, as our findings require further mechanistic studies. Second, the lack of a group with healthy obese individuals limits the understanding of the obesity paradox in this specific context, as we were unable to explore the role of obesity alone. Third, despite our groups being balanced in terms of potential confounders, systolic and diastolic pressure was observed to be higher in obese patients with heart failure in comparison to lean heart failure participants. Curiously, the study cohort presented more prevalence of hypertension in lean HF patients. This paradoxical difference may have occurred because obese HF patients, when they have hypertension, present with higher systolic and diastolic pressures, while, despite hypertension being more prevalent, lean HF patients have lower systolic and diastolic pressures [25,26]. In addition, it is possible that other remaining confounding characteristics are present in these populations that were not observed during data acquisition.

It is also important to mention that when this project was conceived, the main goal was to assess miRNAs with a potential use as biomarkers of the obesity paradox in heart failure. Along the journey, several obstacles were present: (i) There was a lack of a lean control group. (ii) Our results did not present any miRNAs at the intersection of HF–lean vs. HF–obese/HF–lean vs. control/HF–obese vs. control, which would represent DE miRNAs in obese vs. both lean HF patients and the healthy population. (iii) Also, we did not assess whether our cohort presented an obesity paradox or not. Together, these limitations do not allow any inferences in regard to the obesity paradox, but only in regard to miRNAs associated independently with HF or obesity.

Finally, the study’s findings are in a single center, which limits the generalizability of the results to broader and more diverse populations. Our results may be especially relevant for similar populations treated in specialized cardiology centers. However, future studies with larger cohorts from more heterogeneous clinical and socioeconomic backgrounds, multicenter cohorts, and more rigorous design are warranted to confirm these findings and clarify the underlying mechanisms.

## 4. Materials and Methods

### 4.1. Patients and Controls

Patients were recruited from the Heart Failure and Transplant Clinic of our institution between March 2012 and December 2013 for this case–control study. Preliminary data from this cohort had been previously published [13], and biological samples were stored at −80 °C for future analyses. The sample size was determined based on convenience sampling, using the available patients and controls recruited during the study period. No formal sample size calculation was performed due to the exploratory nature of this study.

Eligible participants were individuals diagnosed with stable heart failure (HF), aged between 18 and 80 years, with a left ventricular ejection fraction (LVEF) below 45%. Obese patients had BMI ≥ 30 kg/m^2^ (body weight in kg/height in meters squared) and percent body fat > 25% in men and >35% in women, in accordance with the World Health Organization criteria for obesity [2], measured by a bioelectrical impedance analyzer (Model BIA 450, Tetrapolar, Biodynamics Corporation, WA, USA). Lean subjects were defined by BMI < 25 kg/m^2^, and body fat was ≤22% in men and ≤32% in women [27]. Healthy controls were blood donors, recruited from the Blood Center at our institution or from the Echocardiography Laboratory when a patient showed normal ECG. These individuals had no history of structural heart disease or HF symptoms and were classified as having normal body mass composition.

Exclusion criteria for all participants were cell dyscrasias, active inflammation, malignant disease, and severe hepatic or renal disease (creatinine > 3 mg/dL). Additional exclusion criteria for HF patients consisted of (I) decompensation episodes within 30 days before inclusion, (II) acute coronary syndrome in the preceding three months, and (III) the presence of implantable cardiac devices, such as pacemakers, implantable cardioverter defibrillators, or cardiac resynchronization therapy defibrillators (CRT-D), due to the interference it may cause with bioelectrical impedance analysis.

In accordance with selection criteria, subjects were allocated into three groups: (I) lean healthy controls, (II) lean with HF, and (III) obese with HF. All groups were balanced by age and sex. In addition, HF subject groups were also balanced by LVEF, HF etiology (ischemic and non-ischemic), New York Heart Association (NYHA) classification, use of beta-blockers, and use of angiotensin-converting enzyme (ACE) inhibitor or angiotensin receptor blocker (ARB).

### 4.2. Data Collection

Data were retrospectively analyzed and assessed on 15 January 2020, and stored biological samples were linked to dataset information through participant identification numbers. Collected data included demographic information, clinical data, comorbidities, and echocardiographic, electrocardiographic, and laboratory results. Body mass index was calculated as weight in kilograms divided by squared height in meters (kg/m^2^). Bioimpedance analysis was performed with tetrapolar bioimpedance of Biodynamics, model 450. Waist circumference was measured between the ribs and the iliac crest (after air expiration) by a trained researcher.

### 4.3. Sample Preparation

Blood samples were withdrawn from participants in non-fasting conditions from their peripheral veins into EDTA-coated tubes. Plasma was assessed within one hour after collection by centrifugation at 1500 rpm for 15 min at 4 °C. All samples were aliquoted in 500 μL microtubes and stored at −80 °C for further analysis.

### 4.4. Microarray Analysis

For the microarray, 30 participants were analyzed (10 HF–lean, 10 HF–obese, and 10 controls), and total RNA was extracted from 396 μL of plasma spiked-in with 4 μL of cel-miR-39-3p (40nM). Then, 1.2 mL of Trizol LS reagent was added and incubated for 5 min. Next, 0.32 mL of chloroform was added, incubated for 15 min, and centrifuged at 12,000× *g* at 4 °C for 15 min. Then, we removed the supernatant, added 0.8 mL of isopropyl alcohol, incubated for 10 min, and centrifuged at 12,000× *g* at 4 °C for 10 min. After removing the isopropyl alcohol, 1.6 mL of 75% ethanol was added and centrifuged at 7500× *g*, at 4 °C, for 5 min. Dry for 10 min and resuspend in 16 uL of RNase-free water, placed in a dry bath at 58 °C for 10 min. The concentration of miRNAs was determined by spectrophotometric analysis (NanoDrop 1000, Thermo Scientific, Wilmington, DE, USA).

The Affymetrix GeneChip miRNA 4.0 Array was used to scan the miRNAs in patient samples. RNA labeling and array hybridization were performed following the manufacturer’s instructions. Differential analysis of the miRNAs was performed using Affymetrix GeneChip miRNA array technology version 4.0 (Thermo Fisher Scientific, Santa Clara, CA, USA), according to the manufacturer’s specifications, on total RNA extracted from plasma. The array contains probesets for 3770 mature human miRNAs. Total RNA (160 ng; 20 ng/μL), including miRNAs, was biotin-labeled using the FlashTagTM Biotin HSR RNA Labeling kit (Affymetrix, Genisphere, Hatfield, PA, USA), and the samples were then hybridized overnight using the GeneChip Hybridization Oven 640 (Affymetrix, Santa Clara, CA, USA) at 48 °C. The arrays were then washed and stained in the GeneChip Fluidics Station 450 (Affymetrix, Santa Clara, CA, USA). The arrays were scanned using a GeneChip Scanner 3000 7 G (Affymetrix, Santa Clara, CA, USA), and the signal values were evaluated using Expression Console Software (EC) v1.2 (Affymetrix by Thermo Fisher Scientific). Intensity values (presence/absence values) and signal histograms of each hybridization were quality-checked.

Raw data were pre-processed using the Robust Multiarray Average (RMA) method, which applies background correction, log2 transformation, and quantile normalization. The batch effect was adjusted with Surrogate Variable Analysis through the sva R package (v. 3.56.0) [28]. Pairwise differential expression analysis was carried out with the Limma R package (v. 3.21) [29]. For identification of differentially expressed miRNAs, we adopted a *p*-value < 0.01 and a log-fold change (logFC) < −1 or >1.

The data used in this publication have been deposited in NCBI’s Gene Expression Omnibus and are accessible through GEO Series accession number GSE288767 (https://www.ncbi.nlm.nih.gov/geo/query/acc.cgi?acc=GSE288767, accessed on 4 February 2025).

### 4.5. RT-qPCR

For miRNA validation, we used plasma from 80 participants (35 HF–lean, 26 HF–obese, and 16 controls). Circulating miRNA from 198 μL of plasma spiked-in with 2 μL of customized 5′-phosphorylated cel-miR-39-3p (40 nM) (Thermo Fisher, #10620310, Waltham, MA, USA) was extracted using the miRNeasy Serum/Plasma Kit. We added 1 mL of QIAzol and incubated for 5 min. Then, 200 μL of chloroform was added, vortexed for 15 s, incubated for 3 min, and centrifuged for 15 min at 12,000× *g* at 4 °C. We removed the supernatant and added 1.5 volumes of 100% ethanol. Afterward, we pipetted 700 μL into the extraction column and centrifuged at 8000× *g* for 60 s at room temperature (15–25 °C). We then added 700 μL of RWT buffer to the column and centrifuged for 60 s at 8000× *g*. Next, we pipetted 500 μL of RPE buffer onto the column and centrifuged for 60 s at 8000× *g*. Then, 500 μL of 80% ethanol was pipetted onto the column and then centrifuged for 2 min at 8000× *g*. Afterward, we centrifuged the column in a new tube for another 5 min to eliminate the rest of the ethanol. In the end, we added 14 μL of RNase-free water and centrifuged at 8000× *g* for 60 s at room temperature (15–25 °C). The concentration of miRNAs was determined by spectrophotometric analysis (NanoDrop 1000).

The Reverse Transcriptase reaction was performed using the TaqManTM Advanced miRNA cDNA Synthesis Kit in StepOnePlus™ equipment (Applied Biosystems, Foster City, CA, USA). For the poly-A tail reaction, 1.75 μL of sample in 2.75 μL of Poly (A) Reaction Mix was added to each tube. Then, the samples were placed into a thermocycler for polyadenylation at a temperature of 37 °C for 45 min, and the reaction was stopped at 65 °C for 10 min. Next, the adapter ligation reaction was performed with 7.5 μL of Ligation Reaction Mix, at a temperature of 16 °C for 60 min. After, reverse transcription (RT) was performed using 11.25 μL of RT Reaction Mix at 42 °C for 15 min, and the reaction was stopped at 85 °C for 5 min. The miR-Amp reaction was performed with 3.75 μL of the RT product in 33.75 μL of the miR-Amp Reaction Mix, activating the enzyme at 95 °C for 5 min, denaturing at 95 °C for 3 s, annealing, and extending at 60 °C for 30 s (14 cycles—denaturation, annealing, and extension phases), and the reaction was stopped at 99 °C for 10 min.

Each amplification reaction was prepared with 5 μL of TaqMan^®^ Fast Advanced Master Mix (2X) (Thermo Fisher, Waltham, MA, USA), 0.5 μL of the TaqMan^®^ Advanced miRNA Assay (20X) (Thermo Fisher, Waltham, MA, USA), and 2 μL of RNase-free water, in the MicroAmp^®^ Flat Optical 48-Well Reaction Plate (0.1 mL) (Thermo Fisher, Waltham, MA, USA); enzyme activation at 95 °C, for 20 s; denaturation at 95 °C for 1 s; and annealing and extension at 60 °C for 20 s (40 cycles—denaturation, annealing, and extension phases).

In brief, data from qRT-PCR were normalized to the reference control miRNA, cel-miR-39-3p, using the ΔCT method. The relative expression levels for each individual miRNA were calculated using the following mathematical formula: ΔCT = CTsample − CTcel-miR-39-3p. The ΔΔCT for each miRNA was calculated using the following formula: ΔΔCT = ΔCTsample − ΔCT mean of control group. These values were transformed into quantities using the formula 2ΔΔCT and are presented as the fold change relative to the internal control.

Researchers responsible for miRNA quantification were blinded to group allocation, and all sample collections were performed according to standardized protocols by trained professionals.

### 4.6. Target and Pathway Analysis of Validated miRNAs

The differentially expressed miRNAs validated through qRT-PCR, hsa-miR-451a, hsa-miR-22-3p, and hsa-miR-548ac, were submitted for target analysis using miRTarBase release 8.0 [30] and TarBase v8.0 [31]. Both databases contain experimentally validated miRNA–target interactions and were explored by applying the following filtering criteria: (I) for miRTarBase, we restricted analysis to interactions classified as functional (both weak and strong); (II) for TarBase, we considered only interactions classified as “positive” and related to a direct association between miRNAs and their target gene. Moreover, for TarBase, all interactions supported by HITS-CLIP and PAR-CLIP high-throughput experiments were further analyzed to count the number of supporting experiments per interaction. To mitigate the impact of publication bias (i.e., the fact that some miRNAs are studied more frequently than others), we applied a uniform selection criterion for all analyzed miRNA–target interactions. Specifically, only interactions with at least 2 supporting pieces of evidence were retained, regardless of the miRNA. This approach was applied to hsa-miR-451a, hsa-miR-22-3p, and hsa-miR-548ac, ensuring consistency in the analysis process.

The union of miRNA–target interactions retrieved from both databases was used as the interactome of the validated miRNAs. The functional enrichment was quantitatively assessed (*p*-value) using a hypergeometric distribution. Multiple test correction was also implemented by applying the False Discovery Rate (FDR) algorithm [32] at a significance level of *p* < 0.05. We used the CluePedia—a ClueGO plugin for pathway insights using integrated experimental and in silico data—v.2.5.7 [33,34], consulting the KEGG database for the terms or pathways [35].

### 4.7. Ethical Considerations

The study protocol was approved by our institution’s review board under the numbers 120084 and 19470. The ethical evaluation was certified with the number CAEE: 241448.1.0000.5327. The protocol was conducted following the Declaration of Helsinki, and all patients signed a written informed consent form prior to their inclusion.

### 4.8. Statistical Analysis

Quantitative data are described as mean and standard deviation and frequencies as absolute values and percentages. In the RT-PCR analysis, the differential expression of miRNAs was analyzed using the geometric mean and standard deviation. For group comparisons, we used the Kruskal–Wallis test and Dunn’s multiple comparison test, with a *p*-value < 0.05.

## 5. Conclusions

In this case–control study, miR-451a, miR-22-3p, and miR-548ac were identified as circulating microRNAs up-regulated in patients with heart failure, regardless of obesity status. These findings suggest that these miRNAs are more closely related to the pathophysiological mechanisms of heart failure than to the obesity paradox. Functional analyses indicated their association with key signaling pathways involved in cardiac remodeling, metabolism, and cell survival, including AKT1 and MAPK1 pathways.

However, due to the limited sample size, the use of convenience sampling, and the absence of a group of healthy obese individuals, these results should be interpreted as an exploratory study. Further studies with larger, multicenter cohorts and longitudinal designs are warranted to confirm these associations and explore their potential role as biomarkers or therapeutic targets in heart failure.

## Figures and Tables

**Figure 1 ijms-26-09475-f001:**
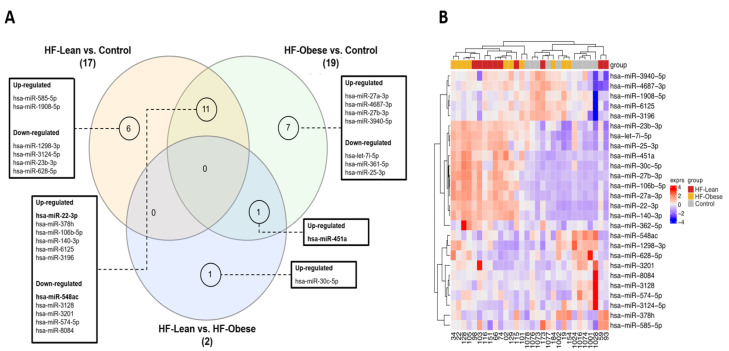
(**A**)**.** Venn diagram of differentially expressed miRNAs. Venn diagram illustrating differentially expressed microRNAs across three comparisons: HF–lean vs. controls, HF–obese vs. controls, and HF–lean vs. HF–obese. The numbers represent unique and shared differentially expressed microRNAs. HF–lean vs. controls had 17, of which 6 were unique. HF–obese vs. controls showed 19, with 7 being unique. HF–lean vs. HF–obese identified 2, with 1 being unique. Eleven microRNAs were commonly differentially expressed in HF–lean and HF–obese compared to controls, while only one was shared between HF–obese vs. control and HF–lean vs. HF–obese. The adjacent table lists up- and down-regulated microRNAs for each comparison. (**B**). Heatmap of differentially expressed miRNAs. Hierarchical clustering was applied to both samples (gray = control; yellow = HF–obese; and red = HF–lean) and miRNA expression levels (down-regulation from dark blue to up-regulation in deep red) using Person correlation similarity and complete linkage.

**Figure 2 ijms-26-09475-f002:**
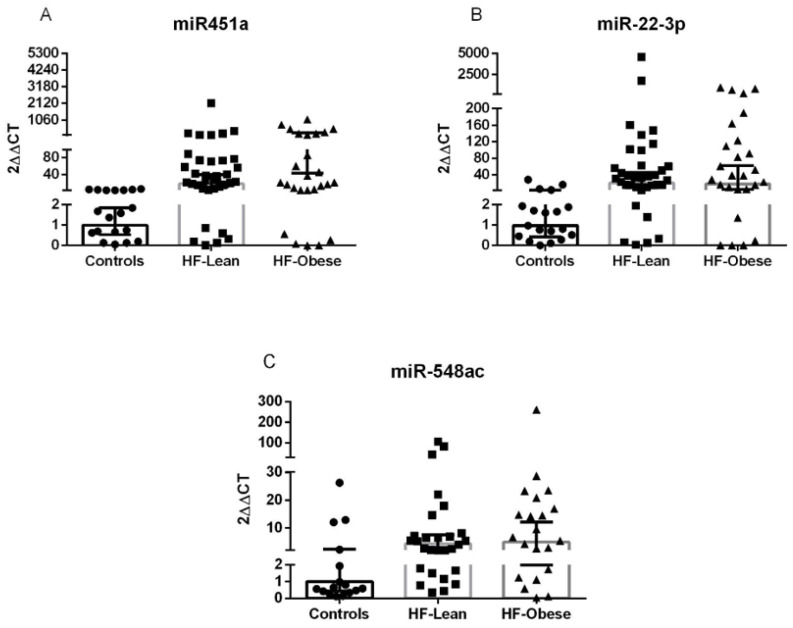
miRNAS validated by qRT-PCR. Validation of selected microRNA expression in experimental groups (●: Controls; ■: HF- Lean; and ▲: HF-obese). The relative expression of microRNAs miR-451a (control = 19; HF–lean = 35; and HF–obese = 26) (**A**), miR-22-3p (control = 19; HF–lean = 35; and HF–obese = 26) (**B**), and miR-548ac (control = 16; HF–lean = 28; and HF–obese = 21) (**C**) was analyzed in the control, HF–lean, and HF–obese groups. The microRNAs were selected based on microarray results. miR-451a and miR-22-3p were up-regulated in the HF groups, validating the microarray, while miR-548ac was not confirmed.

**Figure 3 ijms-26-09475-f003:**
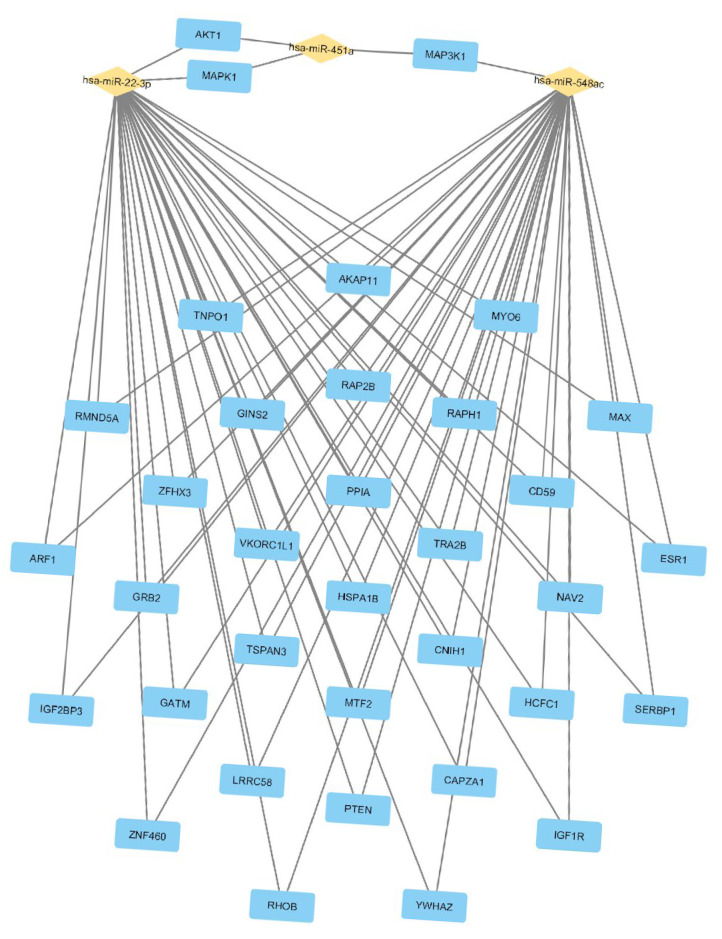
miRNA–gene network. Interaction network between microRNAs and target genes. The figure shows three microRNAs—hsa-miR-22-3p, hsa-miR-451a, and hsa-miR-548ac (in yellow)—and their respective target genes (in blue). The lines connecting the miRNAs to the genes represent regulatory interactions, indicating that these miRNAs control the expression of the linked genes.

**Figure 4 ijms-26-09475-f004:**
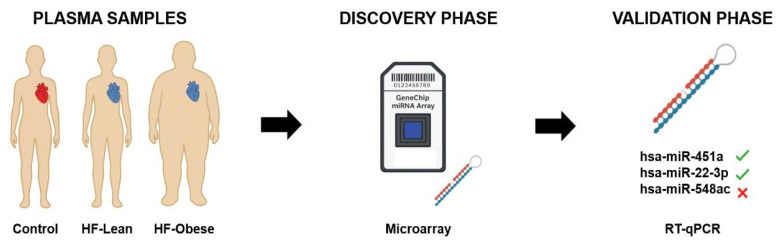
Recapitulative figure. The figure shows the flux of information, from the discovery cohort to the validated miRNAs. Healthy heart is represented in red, while heart failure is represented in blue.

**Table 1 ijms-26-09475-t001:** Discovery cohort baseline characteristics.

Characteristics	Controls (*n* = 10)	HF–Lean (*n* = 10)	HF–Obese (*n* = 10)	*p*-Value
Age (years)	54.09 ± 10.70	53.11 ± 10.87	54.56 ± 15.00	0.965
Male sex (%)	7 (70)	7 (70)	5 (50)	0.563
White ethnicity (%)	9 (90)	6 (60)	7 (70)	0.535
Body weight (kg)	69.80 ± 11.86	60.00 ± 11.44	106.23 ± 24.41 **	0.000004
Body mass index (kg/m^2^)	23.77 ± 2.10	21.62 ± 2.07	38.09 ± 5.91 **	<0.000001
Body fat (%)	15.91 ± 6.98	14.28 ± 5.13	34.92 ± 5.44 **	<0.000001
Waist circumference (cm)		82.82 ± 9.19	119.34 ± 13.23 **	0.000001
Heart rate (beats/min)		77.44 ± 11.95	76.22 ± 12.02	0.831
Echocardiography indices				
LVEF (%)		29.30 ± 7.09	29.20 ± 10.15	0.980
LVEDD (mm)		64.70 ± 11.62	66.30 ± 4.52	0.690
LVESD (mm)		55.80 ± 11.65	56.10 ± 5.90	0.943
NYHA class (%)				
I or II		9 (90)	8 (80)	0.531
III		1 (10)	2 (20)	0.531
HF etiology (%)				
Ischemic		5 (50)	6 (60)	0.653
Non-ischemic		5 (50)	4 (40)	0.653
Medical history (%)				
Hypertension		4 (40)	6 (60)	0.371
Diabetes		3 (30)	3 (30)	1.0
Atrial fibrillation or flutter		2 (20)	2 (20)	1.0
Myocardial Infarction		4 (40)	5 (50)	0.653
Medication (%)				
ACE inhibitor or ARB		9 (90)	10 (100)	0.305
Beta-blocker		6(60)	8 (80)	0.329
Diuretic		8 (80)	9 (90)	0.531
Statins		4 (40)	7 (70)	0.178
Calcium channel blocker		0 (0%)	3 (30)	0.06
Vasodilators		3 (30)	2 (20)	0.606
Nitrates		4 (40)	4 (40)	1.0
Antiarrhythmics		1 (10)	0 (0)	0.305
Digitalis		9 (90)	9 (90)	1.0
Antiplatelet agents		6 (60)	5 (50)	0.653
Oral anticoagulants		3 (30)	0 (0)	0.06
Hypoglycemic		2 (20)	0 (0)	0.136
Blood pressure (mmHg)				
Systolic		125.00 ± 22.77	118.80 ± 20.81	0.533
Diastolic		73.40 ± 12.58	77.10 ± 14.41	0.548
Laboratory				
Creatinine (mg/dL)		1.26 ± 0.48	1.14 ± 0.27	0.492
Hemoglobin (g/dL)		13.40 ± 1.42	13.65 ± 1.73	0.761
Total cholesterol (mg/dL)		150.14 ± 30.71	170.14 ± 27.40	0.223
LDL cholesterol (mg/dL)		67.15± 35.21	77.83 ± 41.08	0.597
HDL cholesterol (mg/dL)		42.29 ± 16.05	38.86 ± 6.82	0.612
Triglycerides (mg/dL)		155.57 ± 95.34	164.33 ± 41.87	0.839
Albumin (g/dL)		4.38 ± 0.24	4.43 ± 0.16	0.671

Table 1 presents the baseline characteristics of the discovery cohort, which includes three groups: control (*n* = 10), heart failure–lean (HF–lean, *n* = 10), and heart failure–obese (HF–obese, *n* = 10). The *p*-values indicate statistically significant differences between the groups, with values less than 0.05 (<0.05). ** Statistically significant difference between the Obese group and Lean.

**Table 2 ijms-26-09475-t002:** Validation cohort baseline characteristics.

Characteristics	Controls (*n* = 19)	HF–Lean (*n* = 35)	HF–Obese (*n* = 26)	*p*-Value
Age (years)	49.3 ± 12.43	56.22 ± 10.54	56.50 ± 13.44	0.089
Male sex (%)	12 (63.2)	22 (62.9)	17 (65.4)	0.978
White ethnicity (%)	17 (89.5)	24 (68.6)	17 (68.0)	0.258
Body weight (kg)	69.29 ± 10.98	60.32 ± 9.09	100.32 ± 18.65 *	<0.001
Body mass index (kg/m^2^)	24.40 ± 2.51	21.74 ± 2.06 *	36.98 ± 4.96 *	<0.001
Body fat (%)	15.88 ± 5.68	16.94 ± 6.35	32.94 ± 5.19 *	<0.001
Waist circumference (cm)		84.57 ± 7.84	116.73 ± 11.04	<0.001
Heart rate (beats/min)		73.15 ± 14.21	80.08 ± 13.95	0.069
Echocardiography indices				
LVEF (%)		31.85 ± 10.25	30.04 ± 8.85	0.474
LVEDD (mm)		63.68 ± 9.44	65.50 ± 6.77	0.408
LVESD (mm)		51.48 ± 13.68	55.62 ± 7.43	0.171
NYHA class (%)				
I or II		30 (85.7)	21 (80.8)	0.606
III		5 (14.3)	5 (19.2)	0.606
HF etiology (%)				
Ischemic		12 (34.3)	8 (30.8)	0.772
Non-ischemic		23 (65.7)	18 (69.2)	0.772
Medical history (%)				
Hypertension		13 (37.1)	7 (26.9)	0.05
Diabetes		9 (25.7)	9 (34.6)	0.451
Atrial fibrillation or flutter		8 (22.9)	10 (38.5)	0.186
Myocardial Infarction		9 (25.7)	7 (26.9)	0.915
Medication (%)				
ACE inhibitor or ARB		34 (97.1)	25 (96.2)	0.830
Beta-blocker		27 (77.1)	20 (76.9)	0.984
Diuretic		28 (80)	25 (96.2)	0.065
Statins		14 (40)	13 (50)	0.437
Calcium channel blocker		0 (0)	4 (15.4)	0.016
Vasodilators		6 (17.1)	8 (30.8)	0.211
Nitrates		9 (25.7)	7 (26.9)	0.915
Antiarrhythmics		3 (8.6)	1 (3.8)	0.461
Digitalis		30 (85.7)	22 (42.3)	0.905
Antiplatelet agents		16 (45.7)	10 (38.5)	0.571
Oral anticoagulants		9 (25.7)	4 (30.8)	0.330
Hypoglycemic		4 (11.4)	4 (15.4)	0.651
Blood pressure (mmHg)				
Systolic		116.69 ± 23.95	136.96 ± 31.20	0.006
Diastolic		70.46 ± 14.58	84.69 ± 14.14	<0.001
Laboratory				
Creatinine (mg/dL)		1.18 ± 0.41	1.12 ± 0.36	0.609
Hemoglobin (g/dL)		13.65 ± 1.57	56.27 ± 205.51	0.252
Total cholesterol (mg/dL)		172.81 ± 47.09	180.42 ± 92.89	0.717
LDL cholesterol (mg/dL)		93.07 ± 40.18	82.34 ± 8.03	0.361
HDL cholesterol (mg/dL)		42.70 ± 11.94	39.11 ± 10.61	0.302
Triglycerides (mg/dL)		210.56 ± 316.25	375.22 ± 954.58	0.409
Albumin (g/dL)		4.34 ± 0.26	4.44 ± 0.21	0.279

Table 2: Baseline characteristics of the validation cohort. This table presents the baseline characteristics of the three groups in the validation cohort: controls (*n* = 19), HF–lean (*n* = 35), and HF–obese (*n* = 26). Values are presented as mean ± standard deviation. The *p*-value column indicates statistically significant differences between the groups. * Difference between HF groups and control, *p*-value < 0.05.

## Data Availability

Anonymized participants’ clinical information and qPCR results are fully available in the Open Science Framework (OSF) (DOI 10.17605/OSF.IO/9BQSR). Microarray data are available in the Global Expression Omnibus (GEO) under Series accession number GSE288767 (https://www.ncbi.nlm.nih.gov/geo/query/acc.cgi?acc=GSE288767, accessed on 4 February 2025). In addition, we have included as Appendix A the data of differentially expressed results for the microarray and the correlation of microarray vs. qPCR results of the validated miRNAs.

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
