# Peer review of "A Microarray, Validation, and Gene-Enrichment Approach for Assessing Differentially Expressed Circulating miRNAs in Obese and Lean Heart Failure Patients: A Case–Control Study"

_ijms, 2025, doi:10.3390/ijms26199475_

Round 1

Reviewer 1 Report

Comments and Suggestions for Authors

This manuscript had obtained some interesting clinical findings and some potential biomarkers for HF had been identified and validated. However, since the main objective of this study was to identify and validate differentially expressed circulating miRNAs in HF-obese and HF-lean patients, why the authors did not validate the changes in mir-30c-5p? They should focus on the difference between the two groups. It is also confusing that only two miRNAs were identified in comparation the HF-obese and HF-lean groups. Do the authors have some explanations? Whether the sensitivity of microarray was good enough?

Author Response

General comment: This manuscript had obtained some interesting clinical findings and some potential biomarkers for HF had been identified and validated. However, since the main objective of this study was to identify and validate differentially expressed circulating miRNAs in HF-obese and HF-lean patients.

Comment 1: Why did the authors not validate the changes in mir-30c-5p? They should focus on the difference between the two groups. It is also confusing that only two miRNAs were identified in comparison the HF-obese and HF-lean groups. Do the authors have some explanations? 

Response 1: We thank the reviewer for this thoughtful comment. Regarding miR-30c-5p, although it was identified in the comparison between Lean HF and Obese HF groups, it did not show significant differences when compared with the control group. This suggests that the observed variation is more closely related to obesity alone, rather than to the interaction between heart failure and obesity.

Since our aim was to identify markers specifically associated with the interplay between HF and obesity, we chose to focus on the intersection of two comparisons (Lean HF vs. Obese HF and Obese HF vs. Control). In this context, miR-451a emerged as the most relevant candidate, as it was differentially expressed in both contrasts.

We acknowledge that not all differentially expressed miRNAs identified in the microarray were validated in our study. However, to ensure transparency and to support future investigations, we have made the complete microarray dataset and cohort dataset publicly available in data repositories. We hope this will enable other research groups to further explore these findings, validate additional miRNAs, and generate new hypotheses based on our exploratory work.

Finally, we acknowledge that financial and logistical limitations constrained the number of miRNAs we were able to validate. Within these constraints, we prioritized those with the strongest biological relevance to the interaction between HF and obesity. We remain hopeful that the availability of our full dataset will enable further validation and exploration by other groups.

Comment 2: Whether the sensitivity of the microarray was good enough?

Response 2: The microarray experiment was conducted by an experienced researcher and was remotely monitored by a field specialist from Thermo Fisher. According to their assessment, the experiments were performed successfully and the platform functioned as expected.

Reviewer 2 Report

Comments and Suggestions for Authors

Thank you for the opportunity to review the manuscript « Microarray, validation and gene-enrichment approach for as- 2 sessing differentially expressed circulating miRNAs in obese 3 and lean heart failure patients: a case-control study».
Obesity is a major risk factors leads to the development of cardiovascular disease and cardiovascular disease mortality independently of other cardiovascular risk factors. Obesity leads to structural and functional changes of the heart, which causes heart failure.   Patients with severe obesity are a very high-risk population in our clinical practice. However, more recent data highlight shows in HF, overweight and obese patients have longer survival than underweight patients, a phenomenon known as the obesity paradox.
In this regard, it is important to enhance research in these areas. And this manuscript is certainly interesting and important.
It should be acknowledged article is colorfull designed and supplied with beautiful illustrative material.
However, the work raises many questions, there are some critical comments to the article from my part:
1. The most obvious question is whether it is possible to obtain correct conclusions, especially in matters concerning the search for genetic markers and predictors  due to the limited sample size? In my opinion, the analysis is not valid in this context and leads to weak conclusions and a number of errors.
2. In my opinion, it is necessary to clearly indicate the design and stages of the study.  Wich part was retrospective? - this question arose on the basis of the data presented on line 250 (Data were retrospectively analyzed..). And when the clinical data were assessed (line 252, etc.)?
3. Also noteworthy is the lack of data on the exact number of patients studied in the section "4. Materials and Methods".
4. Regarding the control groups, perhaps a comparison with the control groups "lean  without HF" and "obese without HF" would be more correct.
5. Table 1: there are no an important clinical data of control group  (Waist Circ., HR, Echo ...), perhaps the reason is incorrect comparing a single control group vs both study groups.
6. The the cited references can be more recent and relevant. There is a lot of new and available publications in this field,  the newest represented is dated 2022. There are a lot of more relevant and modern sources.                                                                                                                       In addition to correcting the above comments, it makes sense to expand the study groups in order to make the results more convincing and valuable.

Author Response

General comment: 

Thank you for the opportunity to review the manuscript « Microarray, validation and gene-enrichment approach for assessing differentially expressed circulating miRNAs in obese and lean heart failure patients: a case-control study».

Obesity is a major risk factor that leads to the development of cardiovascular disease and cardiovascular disease mortality independently of other cardiovascular risk factors. Obesity leads to structural and functional changes of the heart, which causes heart failure.   Patients with severe obesity are a very high-risk population in our clinical practice. However, more recent data highlight shows in HF, overweight and obese patients have longer survival than underweight patients, a phenomenon known as the obesity paradox.

In this regard, it is important to enhance research in these areas. And this manuscript is certainly interesting and important.

It should be acknowledged that the article is colorfully designed and supplied with beautiful illustrative material.

However, the work raises many questions, there are some critical comments to the article from my part.

Comment 1: The most obvious question is whether it is possible to obtain correct conclusions, especially in matters concerning the search for genetic markers and predictors  due to the limited sample size? In my opinion, the analysis is not valid in this context and leads to weak conclusions and a number of errors.

Response 1: We thank the reviewer for raising this important point regarding sample size. We partially agree with the reviewer’s concern. Our study was conducted in two stages: in the first stage, circulating microRNAs were explored using microarray technology in 30 participants (10 controls and 20 heart failure patients — 10 lean and 10 obese). In the second stage, selected findings were validated in a larger cohort of 80 participants (19 controls, 35 lean HF, and 26 obese HF).

It is noteworthy that most publications in the field of circulating miRNAs report similar sample sizes, with an average of approximately 50 participants, as summarized in the table provided. This indicates that our study is consistent with existing literature.

id_article

DOI

n_control

n_case

1a

10.1253/circj.CJ-10-0457

17

17

2

10.1097/MD.0000000000017710

245

245

3b

10.1159/000493419

30

80

4a

10.1097/MD.0000000000021018

215

91

4b

10.1097/MD.0000000000021018

124

5

10.1210/jc.2018-00684

80

80

6

10.1039/c6mb00596a

22

35

7

10.1111/jcmm.13813

14

30

8a

10.1590/1414-431X20154590

32

32

8b

10.1590/1414-431X20154590

36

9

10.1371/journal.pone.0204235

10

10

10

10.1002/ehf2.12597

21

39

12

10.5114/aoms.2019.82919

10

13

13

10.1093/eurjhf/hft078

15

44

14b

10.5603/CJ.a2016.0097

20

19

15b

10.1016/j.hjc.2017.10.002

20

33

16b

10.1002/ejhf.223

30

60

17a

10.1016/j.gene.2015.07.068

17

20

17b

10.1016/j.gene.2015.07.068

17

20

18

10.1016/j.ihj.2012.12.022

39

45

19a

10.1016/j.healun.2017.02.008

15

25

19b

10.1016/j.healun.2017.02.008

29

20a

10.1016/j.orcp.2015.01.006

41

40

20b

10.1016/j.orcp.2015.01.006

40

21

10.1093/eurjhf/hft088

15

81

24b

10.1016/j.ijcard.2016.07.179

12

32

25

10.1038/srep33580

41

50

26b

10.1016/j.ncrna.2020.08.001

60

60

27

10.1080/13813455.2020.1775655

13

29

28a

10.3892/mmr.2017.7575

40

80

29

10.1016/j.ijcard.2011.11.082

10

41

31

10.1042/BSR20191653

62

62

32

10.12659/MSM.911632

31

94

33

10.1016/j.jacc.2018.11.060

208

338

34a

10.1073/pnas.1401724111

13

14

34b

10.1073/pnas.1401724111

24

35

10.3390/ijms21249509

25

78

36b

10.3892/mmr.2018.9734

10

10

39a

10.1111/eci.13308

100

100

39b

10.1111/eci.13308

100

41

10.1111/jcmm.15306

10

8

43

10.26355/eurrev_202001_19937

60

60

44a

10.1210/jc.2013-2218

6

6

44b

10.1210/jc.2013-2218

6

45b

10.1530/EJE-14-0867

107

123

46

10.7150/ijms.18988

24

17

47

10.1177/0300060519882583

80

90

48

10.3892/etm.2019.8318

30

50

49a

10.1016/j.ijcard.2013.01.160

30

55

49b

10.1016/j.ijcard.2013.01.160

51

50a

10.1016/j.rec.2020.08.012

76

60

50b

10.1016/j.rec.2020.08.012

32

51b

10.18632/oncotarget.6631

45

45

52a

10.1186/s13104-017-3090-y

20

40

52b

10.1186/s13104-017-3090-y

53

10.1159/000370028

78

78

54

10.3892/etm.2020.8599

60

92

55

10. 1136/bmjdrc- 2020- 001441

47

155

58

10.1038/srep40696

15

13

59

10.1007/s12265-018-9858-1

35

35

61b

10.1371/journal.pone.0136404

10

18

62b

10.1002/jcla.23246

-

600

63b

10.1002/ehf2.13090

85

71

64a

10.3390/ijms17101620

30

62

64b

10.3390/ijms17101620

30

65b

10.1161/CIRCRESAHA.110.218297

39

30

67a

10.1002/oby.21950

12

11

67b

10.1002/oby.21950

19

67c

10.1002/oby.21950

15

68b

10.1093/eurheartj/eht256

8

14

69

10.1016/j.trsl.2020.01.003

70

30

70a

10.111/apm.12389

25

25

71

10.1016/j.amjcard.2013.11.060

35

41

72a

10.1371/journal.pone.0077251

20

20

72b

10.1371/journal.pone.0077251

16

73

10.1093/eurjhf/hfr155

30

30

75

10.1016/j.carpath.2013.04.001

22

18

76a

10.2147/DMSO.S262888

43

34

76b

10.2147/DMSO.S262888

21

79

10.1373/clinchem.2012.195776

49

19

80

10.1089/gtmb.2020.0034

30

30

81a

10.1089/gymb.2018.0188

10

25

81b

10.1089/gymb.2018.0188

25

84

10.1002/ejhf.517

8

9

85

10.1002/ejhf.119

11

64

86

10.3109/135475oX.2013.870605

10

17

87a

10.1159/000476029

60

35

87b

10.1159/000476029

48

35

Mean

41.2

54.9

48.1

SD

45.2

76.5

64.5

We acknowledge that the limited sample size reduces statistical power and may increase the risk of type II errors. We have clearly stated this limitation in the Discussion (lines 215–217):

“First, no formal sample size calculation was performed, the sample size was relatively small and based on convenience sampling, which may limit the statistical power to detect subtle differences and increase the risk of type II error.”

To promote transparency and facilitate further research, we have made the complete microarray dataset publicly available in GEO (https://www.ncbi.nlm.nih.gov/geo/query/acc.cgi?acc=GSE288767) and the miRNA expression and clinical data from our cohort accessible via the Open Science Framework (DOI 10.17605/OSF.IO/9BQSR). This allows other researchers to build upon our findings and contribute to the growing evidence base.

Finally, our conclusions were framed carefully to reflect both the significance of our findings and the limitations of the study. As noted in the manuscript (lines 434–436):

“In this case-control study, miR-451a, miR-22-3p, and miR-548ac were identified as circulating microRNAs upregulated in patients with heart failure, regardless of obesity status. These findings suggest that these miRNAs are more closely related to the pathophysiological mechanisms of heart failure than to the obesity paradox. Functional analyses indicated their association with key signaling pathways involved in cardiac remodeling, metabolism, and cell survival, including AKT1 and MAPK1 pathways.

However, due to the limited sample size, the use of convenience sampling, and the absence of a group of healthy obese individuals, these results should be interpreted as an exploratory study. Further studies with larger, multicenter cohorts and longitudinal designs are warranted to confirm these associations and explore their potential role as biomarkers or therapeutic targets in heart failure."

We believe that, despite the sample size limitations, our study provides valuable exploratory insights and a transparent foundation for future research in this area.

Comment 2: In my opinion, it is necessary to clearly indicate the design and stages of the study.  Which part was retrospective? - this question arose on the basis of the data presented on line 250 (Data were retrospectively analyzed..). And when the clinical data were assessed (line 252, etc.)?

Response 2: We thank the reviewer for this comment and the opportunity to clarify the study design. This is a case-control study aimed at assessing circulating microRNA expression in patients with heart failure (lean and obese) compared with a control group. For the case arm, we used frozen plasma samples from patients recruited at our institutional heart failure and transplant clinic between March 2012 and December 2013, as part of a previous study from our group (10.1016/j.gene.2015.07.068). For the present work, clinical records of these participants were reviewed and assessed retrospectively in January 2020.

Thus, this study can be considered a case-control study nested within a retrospective cohort. In summary, all microarray experiments, miRNA expression analyses, and in silico analyses were conducted specifically for this study, while the clinical data were retrospectively obtained from existing records.

Comment 3: Also noteworthy is the lack of data on the exact number of patients studied in section "4. Materials and Methods".

Response 3: 

We thank the reviewer for this comment. The exact number of participants for each analysis was already provided in the Results section: in the Discovery cohort subsection (baseline characteristics) and Table 1, and in the Validation cohort subsection (baseline characteristics) and Table 2. Specifically:

  • In lines 61 and 62: “A total of 20 HF patients (10 obese and 10 lean) and 10 healthy controls were en-rolled for the discovery phase.”
  • Table 1 headings
  • In lines 105 and 106:  “A total of 80 subjects, 61 HF patients (26 obese and 35 lean), and 19 healthy controls were enrolled for the validation phase of the study.”
  • Table 2 headings

To facilitate reader access to sample size information, we have now also included it in the Materials and Methods section:

  • In lines 305 and 306: For microarray, 30 participants were analyzed (10 HF-Lean, 10 HF-Obese, and 10 controls),
  • In lines 343 and 344: For miRNA validation, we have used plasma from 80 participants (35 HF-Lean, 26 HF-Obeses, and 16 controls).

Additionally, we have included the sample size for each group in the Figure 2 caption (lines 132–136), ensuring clarity for readers:

Figure 2.  miRNAS validated by qRT-PCR. Validation of selected microRNA expression in experimental groups. The relative expression of microRNAs miR-451a (Control=19, HF-Lean=35, and HF-Obese=26) (A), miR-22-3p (Control=19, HF-Lean=35, and HF-Obese=26) (B), and miR-548ac (Control=16, HF-Lean=28, and HF-Obese=21) (C) was analyzed in the Control, HF-Lean, and HF-Obese groups. The microRNAs were selected based on microarray results. miR-451a and miR-22-3p were upregulated in the HF groups, validating the microarray, while miR-548ac was not confirmed.”

These changes should make the sample size for each analysis clear and easily accessible throughout the manuscript.

Comment 4: Regarding the control groups, perhaps a comparison with the control groups "lean  without HF" and "obese without HF" would be more correct.

Response 4: 

We thank the reviewer for this important observation. We agree that including both lean and obese control groups without heart failure would provide stronger inferences regarding the obesity paradox. As noted in the manuscript, this limitation has been explicitly acknowledged in both the Discussion and Conclusion sections:

In lines 219 to 221:Second, the lack of a group with healthy obese individuals limits the understanding of the obesity paradox in this specific context, as we were unable to explore the role of obesity alone.”

In lines 434 to 436: However, due to the limited sample size, the use of convenience sampling, and the absence of a group of healthy obese individuals, these results should be interpreted as an exploratory study.

We believe that highlighting this limitation ensures transparency and appropriately frames the scope of our findings.

Comment 5: Table 1: there are no important clinical data of control groups  (Waist Circ., HR, Echo ...), perhaps the reason is incorrect comparing a single control group vs both study groups.

Response 5: 

We thank the reviewer for this comment. The primary aim of our study was to assess differentially expressed (DE) microRNAs in a case-control design comparing HF Lean and HF Obese patients versus controls. Once the DE microRNAs were identified, it became possible to explore, as a secondary analysis, correlations between these microRNAs and clinical variables within the HF groups.

Comparing clinical variables between HF patients and controls was not part of the study design; therefore, no clinical measurements (e.g., waist circumference, heart rate, echocardiographic parameters) were collected from control participants. However, some relevant data, such as bioimpedance and BMI, are available in the dataset we have shared via OSF (https://osf.io/dp6v8). We encourage the scientific community to explore this publicly available data for further analyses and potential new insights.

Comment 6: The cited references can be more recent and relevant. There are a lot of new and available publications in this field,  the newest represented is dated 2022. There are a lot of more relevant and modern sources.

Response 6: We thank the reviewer for this valuable suggestion. In response, we have added more recent and relevant references to strengthen the manuscript and enrich the discussion. For example, we have included the following statement in lines 181–188:

In addition, downregulation of miR-451a in the left atrium of patients with recurrent atrial fibrillation after ablation is associated with a higher percentage of atrial fibrosis and worse prognosis [17], reinforcing the cardioprotective role of this miRNA by modulating pathways related to cardiac remodeling and inflammation. Suppression of miR-144/451 bicistronic gene locus play important role in the control of oxidative stress and erythrocyte production, which can have a crucial impact on the progression of heart damage to heart failure, and also in the worsening of heart failure symptoms [18].”

These updates provide a more current perspective and highlight the relevance of our findings within the latest literature.

Comment 7: In addition to correcting the above comments, it makes sense to expand the study groups in order to make the results more convincing and valuable.

Response 7: We thank the reviewer for this suggestion. We acknowledge that expanding the study groups would strengthen the results; however, this work was conducted as part of a PhD thesis, and our research group currently does not have personnel available to continue this study.

We also wish to highlight some of the practical challenges of conducting research in Brazil. Despite the generous support of funding institutions such as FAPERGS and CAPES, significant financial limitations remain, especially when compared with wealthier nations. Moreover, the devaluation of the Brazilian currency (Real) has substantially increased the cost of reagents, which are often priced in US dollars. Considering both personnel and financial constraints, we are unable to expand the study at this time.

Reviewer 3 Report

Comments and Suggestions for Authors

1. The abstract is currently misleading. It states that "MiRNAs -451a, -22-3p, and -548ac were up-regulated in HF-lean and HF-obese groups compared to control". This is inaccurate. The results clearly show that miR-548ac was down-regulated in the microarray discovery phase and only showed up-regulation in the validation phase, meaning the initial result was not validated. The abstract must be revised to accurately reflect this important discrepancy. Failure to do so misrepresents the study's findings.

2. In the validation cohort (Table 2), both systolic and diastolic blood pressure were significantly higher in the HF-Obese group compared to the HF-Lean group (p=0.006 and p<0.001, respectively) , and a trend was noted for a higher prevalence of hypertension (p=0.05). While the text mentions the difference in hypertension, the potential confounding effect of blood pressure itself on miRNA expression is not discussed in the limitations section. Please add a brief comment on this.

3. The primary objective was to investigate miRNAs involved in the "obesity paradox". The conclusion is that the identified miRNAs are more likely related to HF itself, independent of obesity. This is a valid negative finding. The manuscript would be strengthened by making this link between the initial hypothesis and the final conclusion more explicit in the Discussion section.

Author Response

Comment 1: The abstract is currently misleading. It states that "MiRNAs -451a, -22-3p, and -548ac were up-regulated in HF-lean and HF-obese groups compared to control". This is inaccurate. The results clearly show that miR-548ac was down-regulated in the microarray discovery phase and only showed up-regulation in the validation phase, meaning the initial result was not validated. The abstract must be revised to accurately reflect this important discrepancy. Failure to do so misrepresents the study's findings.

 Response 1: 

We thank the reviewer for this important observation. We agree that the abstract should accurately reflect the results from both the discovery and validation phases. However, in accordance with editorial guidelines, the abstract must remain concise (maximum 200 words in a single paragraph), which requires summarizing the findings.

To address the reviewer’s concern, we have updated the abstract to better reflect the results while adhering to the word limit. The revised abstract (lines 19–35) now reads:

Obesity is a risk factor associated with cardiovascular diseases that may lead to heart failure (HF). However, in HF, overweight and obese patients have longer survival than underweight patients, a phenomenon known as the obesity paradox. MiRNAs play a fundamental role in gene regulation involved in obesity and HF. The main objective of this study was to identify and validate differentially expressed circulating miRNAs in HF-obese and HF-lean patients. This case-control study was carried out in two phases; the discovery and validation. In the discovery phase, plasma samples from 20 HF patients and from 10 healthy controls were analyzed using the miRNA 4.0 Affymetrix GeneChip array. Differentially expressed miRNAs were ranked and selected for validation. In this phase, plasma miRNAs miRNAs -451a, -22-3p, and -548ac from 80 patients and controls were analyzed qPCR. Target analysis and functional enrichment analysis were performed. When comparing HF-lean and HF-obese groups compared to control, miRNAs -451a and -22-3p were up-regulated in both discovery and validation phases, while -548ac were down-regulated in the discovery phase and up-regulated in the validation phase, indicating that miRNA changes are independent of obesity. These miRNAs regulate genes and different biological processes associated with metabolic, morphological, and functional outcomes.”

We also emphasize that the discovery phase is exploratory, while the validation phase, conducted in a larger cohort, provides results with higher reliability and confidence. Therefore, reporting the up- or down-regulation of miRNAs in the validation phase better reflects their true biological direction.

Comment 2: In the validation cohort (Table 2), both systolic and diastolic blood pressure were significantly higher in the HF-Obese group compared to the HF-Lean group (p=0.006 and p<0.001, respectively) , and a trend was noted for a higher prevalence of hypertension (p=0.05). While the text mentions the difference in hypertension, the potential confounding effect of blood pressure itself on miRNA expression is not discussed in the limitations section. Please add a brief comment on this.

Response 2: 

We thank the reviewer for this important observation. We have added this limitation to the Discussion (lines 221–230) as follows:

“Third, despite our groups being balanced to potential confounders, it was observed to be higher in the systolic and diastolic pressure in patients obese with heart failure in comparison to lean heart failure participants. Curiously, the study cohort presented more prevalence of hypertension in lean HF patients. This paradoxically difference may be occurring because obese HF patients, when they have hypertension, are presented with higher systolic and diastolic pressures, while, despite more prevalent, lean HF patients have lower systolic and diastolic pressures. In addition, it is possible other remaining confounding characteristics are present in these populations that were not observed during data acquisition.”

This addition clarifies the potential influence of blood pressure differences on miRNA expression and ensures transparency in the interpretation of our findings.

Comment 3: The primary objective was to investigate miRNAs involved in the "obesity paradox". The conclusion is that the identified miRNAs are more likely related to HF itself, independent of obesity. This is a valid negative finding. The manuscript would be strengthened by making this link between the initial hypothesis and the final conclusion more explicit in the Discussion section.

Response 3: 

We thank the reviewer for this insightful comment. We have included a paragraph in the Discussion to explicitly link the initial hypothesis to the study’s conclusions (lines 231–239):

It is also important to mention that when this project was conceived, the main goal was to assess miRNAs with a potential use as biomarkers of the obesity paradox in heart failure. Along the journey, several obstacles were presented: (i) the lack of a lean-control group, then (ii) our results did not present any miRNA in the intersection of HF-lean vs. HF-obese/HF-lean vs. control/HF-obese vs. control, which would represent a DE miRNA in both obese vs lean HF patients and versus the healthy population. (iii) Also, we did not assess whether our cohort presented an obesity paradox or not. Together, these limitations do not allow any inferences in regards to the obesity paradox, but only to miRNAs associated independently with HF or obesity.”

This addition clarifies the rationale and contextualizes our negative findings regarding the obesity paradox, while emphasizing the value of the identified miRNAs in heart failure research.